# Quantum-disordered state of magnetic and electric dipoles in an organic Mott system

M. Shimozawa[1], K. Hashimoto [2], A. Ueda[1], Y. Suzuki[1], K. Sugii[1], S. Yamada[1], Y. Imai[1], R. Kobayashi[2], K. Itoh [2], S. Iguchi[2], M. Naka[3,4], S. Ishihara[3], H. Mori[1], T. Sasaki[2] & M. Yamashita[1]

Strongly enhanced quantum fluctuations often lead to a rich variety of quantum-disordered states. Developing approaches to enhance quantum fluctuations may open paths to realize even more fascinating quantum states. Here, we demonstrate that a coupling of localized spins with the zero-point motion of hydrogen atoms, that is, proton fluctuations in a hydrogen-bonded organic Mott insulator provides a different class of quantum spin liquids (QSLs). We find that divergent dielectric behavior associated with the approach to hydrogen-bond order is suppressed by the quantum proton fluctuations, resulting in a quantum paraelectric (QPE) state. Furthermore, our thermal-transport measurements reveal that a QSL state with gapless spin excitations rapidly emerges upon entering the QPE state. These findings indicate that the quantum proton fluctuations give rise to a QSL—a quantum-disordered state of magnetic and electric dipoles—through the coupling between the electron and proton degrees of freedom.

[1] The Institute for Solid State Physics, The University of Tokyo, Kashiwa, Chiba 277-8581, Japan. [2] Institute for Materials Research, Tohoku University, Aoba-ku, Sendai 980-8577, Japan. [3] Department of Physics, Tohoku University, Sendai 980-8578, Japan. [4] Present address: Waseda Institute for Advanced Study, Waseda University, Shinjuku, Tokyo 169-8050, Japan. M. Shimozawa and K. Hashimoto contributed equally to this work. Correspondence and requests for materials should be addressed to M.S. (email: shimo@issp.u-tokyo.ac.jp) or to K.H. (email: hashimoto@imr.tohoku.ac.jp)

The nature of QSLs has been well established in one-dimensional (1D) spin systems. However, it still remains unclear how QSLs emerge in dimensions greater than one. The celebrated resonating-valence-bond theory on a 2D triangular lattice[1,2] puts forward the possibility that geometrical frustration plays an important role in stabilizing QSLs. In fact, a few candidate materials hosting QSLs have now been reported in materials with 2D triangular lattices[3–8]. Nevertheless, according to subsequent theoretical studies[9,10], the effect of geometrical frustration in the triangular lattice is insufficient to stabilize QSLs, leading to a number of proposed mechanisms that may stabilize the QSL states found in the candidate materials[11]. One of the most promising approaches is to utilize a coupling of spins with charges and orbitals; the former has been discussed near a Mott-insulator-to-metal transition where the charge degrees of freedom begin to delocalize[12–18] and the latter has been considered in the framework of a spin–orbital coupling[19]. Such strategies, however, have been limited to the utilization of internal degrees of freedom of electrons.

The hydrogen-bonded organic Mott insulator $\kappa$-$H_3$(Cat-EDT-TTF)$_2$ (hereafter abbreviated as H-Cat)[5,20–22] may serve as a candidate for a different class of QSLs, where $H_2$Cat-EDT-TTF is catechol-fused ethylenedithiotetrathiafulvalene (see Fig. 1a–c). H-Cat forms a 2D spin-1/2 Heisenberg triangular lattice of Cat-EDT-TTF dimers[20] (Fig. 1e). Despite the antiferromagnetic interaction energy $J/k_B$ of ~80 K, no magnetic order has been observed down to 50 mK; this indicates the realization of a QSL state[5]. A distinct feature of H-Cat is that the 2D $\pi$-electron layers are connected by hydrogen bonds[20,21] (Fig. 1a, c), which is in marked contrast to other 2D organic QSL materials such as $\kappa$-(BEDT-TTF)$_2$Cu$_2$(CN)$_3$ (ref. [3]) and EtMe$_3$Sb[Pd(dmit)$_2$]$_2$ (ref. [4]), where the 2D spin systems are separated by non-magnetic insulating layers. This structural feature of H-Cat is highlighted by deuteration of the hydrogen bonds[21]; specifically, in the deuterated analog of H-Cat, $\kappa$-D$_3$(Cat-EDT-TTF)$_2$ (denoted as D-Cat), deuterium localization occurs at $T_c = 185$ K, accompanied by charge disproportionation within the Cat-EDT-TTF layers, resulting in a non-magnetic ground state (Fig. 1d, f). This demonstrates that the hydrogen bonds in this system strongly couple with the charge and spin degrees of freedom of the $\pi$-electrons in the Cat-EDT-TTF dimers.

In contrast to D-Cat, the hydrogen atoms in H-Cat do not localize down to low temperatures[21]. This is inconsistent with the fact[23] that the potential energy curve of the hydrogen bonds calculated for an isolated supramolecule has a double minimum potential with a large energy barrier of ~800 K (see Fig. 1c, g), which should localize the hydrogen atoms in H-Cat at low temperatures. Recent theoretical calculations[22,23] have pointed out that the potential energy curve has a single-well structure and its bottom becomes very shallow and anharmonic (see Fig. 1g) owing to a many-body effect arising from the network of hydrogen bonds and $\pi$-electrons. In this case, the zero-point motion of the hydrogen atoms (termed "proton fluctuations") can be strongly enhanced by the anharmonic potential curve. In contrast to D-Cat, the enhanced proton fluctuations may delocalize the hydrogen atoms down to absolute zero, providing an opportunity

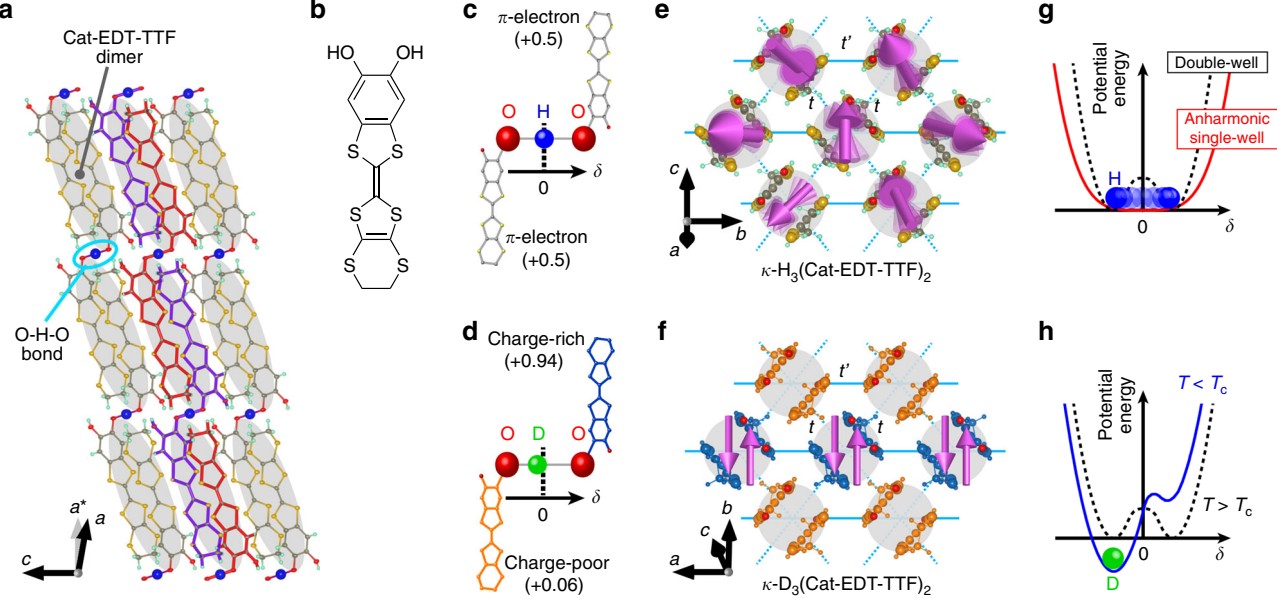

**Fig. 1** Crystal structures of H-Cat and D-Cat. **a** Crystal structure of H-Cat viewed along the *b* axis. **b** Molecular structure of $H_2$Cat-EDT-TTF. **c** Hydrogen-bonded molecular unit $H_3$(Cat-EDT-TTF)$_2$ (supramolecule) in H-Cat. The supramolecules are formed by two [H(Cat-EDT-TTF)]$^{0.5+}$ molecules connected by the hydrogen bond. The supramolecules are stacked along the $(a + c)$ direction, as shown in **a**. For clarity, a part of the stacking columns is colored red and purple in **a**. In the $b$–$c$ plane, two face-to-face [H(Cat-EDT-TTF)]$^{0.5+}$ molecules form a strongly dimerized unit (gray ellipsoid), which generates the 2D $\pi$-electron layers connected by the O–H–O hydrogen bonds (light blue circle). **d** Supramolecular unit in D$_3$(Cat-EDT-TTF)$_2$. Note that $\delta$ in **c**, **d** denotes the displacement of the hydrogen and deuterium atoms from the center of the O–H–O and O–D–O bonds, respectively. **e** Spin and charge structures of the 2D $\pi$-electron layer in H-Cat. The $\pi$-dimers form a slightly anisotropic triangular lattice with $S = 1/2$ spins (magenta arrows). Dotted and solid lines show the inter-dimer hopping integrals, $t$ and $t'$, respectively. **f** Spin and charge structures of a 2D $\pi$-electron layer in D-Cat below the phase transition temperature $T_c$ of 185 K. Charge disproportionation associated with deuterium localization leads to a non-magnetic ground state below $T_c$. The blue- and orange-colored dimers indicate the charge-rich (+0.94) and charge-poor (+0.06) sites, respectively[21]. **g**, **h** Schematics of the potential energy curves of the hydrogen atoms in H-Cat (**g**) and the deuterium atoms in D-Cat (**h**). In H-Cat, the potential energy curve is suggested to change from a double-well structure (dashed line) to a very shallow and anharmonic single-well structure (red solid line) owing to the many-body effect arising from the network of hydrogen bonds and $\pi$-electrons[22,23]. In sharp contrast, the energy curve in D-Cat retains a double-well structure above $T_c = 185$ K (dashed line), leading to the deuterium localization at $T_c$ (blue solid line)

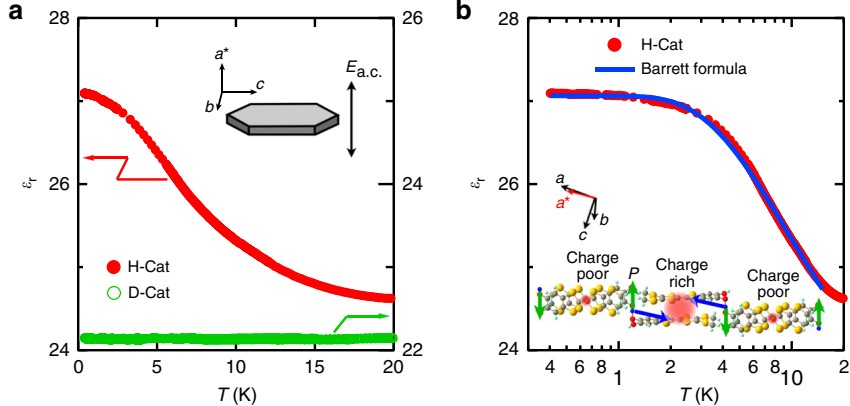

**Fig. 2** Dielectric permittivity of H-Cat and D-Cat. **a** Temperature dependence of the dielectric constant $\epsilon_r(T) = \epsilon/\epsilon_0$ in H-Cat (red, left axis) and D-Cat (green, right axis), where $\epsilon$ is the dielectric permittivity and $\epsilon_0$ is the vacuum permittivity. The inset illustrates our measurement configuration. The dielectric permittivity was measured by applying an a.c. electric field ($E_{a.c.}$) of 1 kV/cm with a frequency of 1 MHz along the $a^\star$ axis. The sample, d.c. bias and frequency dependence of the dielectric behavior is discussed in Supplementary Note 1 (see also Supplementary Fig. 1). **b** The same data for H-Cat plotted on a logarithmic temperature scale. The solid line is a fit to the Barrett formula (see the main text), demonstrating the emergence of a QPE state in H-Cat. The inset shows the electric dipole moments induced within the deuterium bonds and the Cat-EDT-TTF dimers in D-Cat. Because of deuterium localization, the adjacent deuterium bonds possess the local electric dipoles oriented in antiparallel directions (green arrows) below $T_c$. Concomitantly, charge disproportionation is induced within the Cat-EDT-TTF dimers (Fig. 1d), giving rise to the local electric dipoles (blue arrows). Thus, the antiferroelectric interactions are induced by the hydrogen/deuterium-bond dynamics

for realizing a QSL state through strong coupling between the hydrogen bonds and the π-electrons. However, it has not been established whether such strong quantum proton fluctuations are indeed present in H-Cat, and if so, how the quantum fluctuations affect the QSL state.

Here we show, by using a combination of dielectric permittivity and thermal conductivity measurements, that the quantum proton fluctuations presented in H-Cat provide a quantum-disordered state of magnetic and electric dipoles through the coupling between π-electrons and hydrogen atoms. These methods are particularly suitable because the dielectric permittivity is sensitive to local electric-dipole moments arising from hydrogen-bond dynamics[24], whereas the thermal conductivity is a powerful probe to detect itinerant low-lying energy excitations associated with the nature of QSL states[25,26].

## Results

**Dielectric permittivity measurements.** Figure 2 shows the temperature dependence of the dielectric constant $\epsilon_r(T)$ for H-Cat and D-Cat. In H-Cat, $\epsilon_r(T)$ steeply increases with decreasing temperature and then saturates below ~2 K. In sharp contrast, $\epsilon_r(T)$ of D-Cat is temperature-independent owing to deuterium localization (Fig. 2a). The temperature dependence of $\epsilon_r$ for H-Cat is a typical dielectric behavior observed in quantum paraelectric (QPE) materials such as SrTiO$_3$ (ref. [27]), in which long-range electric order is suppressed by strong quantum fluctuations. In the QPE state, $\epsilon_r(T)$ is described by the so-called Barrett formula[28]:

$$\epsilon_r(T) = A + \frac{C}{\frac{T_1}{2} \coth\left(\frac{T_1}{2T}\right) - T_0}. \quad (1)$$

Here, $A$ is a constant offset, $C = n\mu^2/k_B$ is the Curie constant (where $n$ is the density of dipoles, $\mu$ is the local dipole moment, and $k_B$ is the Boltzmann constant), $T_0$ is the Curie–Weiss (CW) temperature in the classical limit (that is, a temperature at which (anti)ferroelectric order occurs in the absence of strong quantum fluctuations) and $T_1$ is the characteristic crossover temperature from the classical CW regime to the QPE regime. As shown in the solid line in Fig. 2b, $\epsilon_r(T)$ of H-Cat is well fitted by the Barrett

formula with $T_0 = -6.4$ K and $T_1 = 7.7$ K; this confirms that strong quantum fluctuations suppress long-range electric order. The relative strength of quantum fluctuations among different QPE materials can be evaluated by the ratio of $T_1$ to $T_0$. The value of $T_1/T_0$ in H-Cat is 1.2, which is smaller than that of the typical QPE material SrTiO$_3$ ($T_1/T_0 = 2.3$, see ref. [27]). This is consistent with the experimental fact that the QPE behavior of SrTiO$_3$ is more significant than that of H-Cat. The obtained negative value of $T_0$ immediately indicates the presence of an antiferroelectric (AFE) interaction in H-Cat, which is consistent with the AFE configuration resulting from deuterium localization in D-Cat (see the inset of Fig. 2b). Therefore, the observed quantum paraelectricity in H-Cat clearly shows that strong quantum fluctuations that suppress the hydrogen-bond order as observed in D-Cat arise from the potential energy curve of H-Cat, consequently leading to the persistence of enhanced proton fluctuations down to low temperatures. The presence of strong quantum fluctuations is consistent with the recent theoretical calculations that highlight the importance of strong many-body effects imposed by the proton–π-electron network on the potential energy curve of H-Cat[22,23].

**Thermal conductivity measurements.** Using thermal conductivity measurements, we next examine how the proton dynamics in the QPE state affects the nature of the QSL state in H-Cat. Figure 3 shows the temperature dependence of the thermal conductivity of H-Cat ($\kappa^H$) and D-Cat ($\kappa^D$). The heat in H-Cat is carried by the phonons ($\kappa_{ph}^H$) and the spin excitations ($\kappa_{sp}^H$), whereas in non-magnetic D-Cat, it is transported only by phonons ($\kappa_{ph}^D$). Assuming that H-Cat and D-Cat share almost identical phonon thermal conductivity ($\kappa_{ph}^H \sim \kappa_{ph}^D$), the relation $\kappa^H = \kappa_{ph}^H + \kappa_{sp}^H \geq \kappa_{ph}^D = \kappa^D$ holds. Unexpectedly, however, we find that $\kappa^H < \kappa^D$ above 2 K (see Fig. 3a), indicating that $\kappa_{ph}^H$ is much more suppressed than $\kappa_{ph}^D$.

To investigate the origin of this suppression, we employ the Callaway model[29], which describes the heat transport of acoustic phonons. Above 2 K, $\kappa^H$ is reproduced by the model including a single resonance scattering mode with a resonance energy of $\hbar\omega_0/k_B \sim 5$–10 K in addition to standard scattering processes (Supplementary Fig. 2; Supplementary Table 1; Supplementary Note 2).

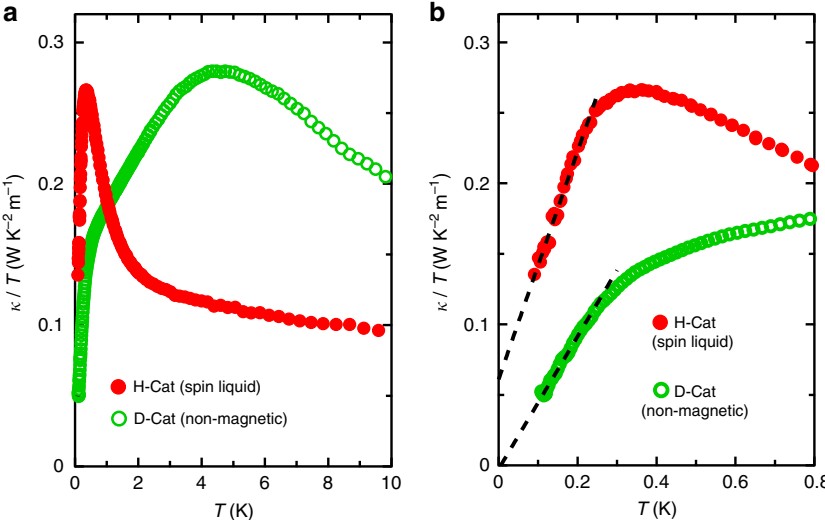

**Fig. 3** Thermal conductivity of H-Cat and D-Cat. **a** Temperature dependence of $\kappa/T$ of H-Cat (filled red symbols) and D-Cat (open green symbols) below 10 K in the zero-field. **b** Low-temperature thermal conductivity for H-Cat and D-Cat below 0.8 K in the zero-field. The extrapolated data (dashed lines) assume that $\kappa_{ph} \propto T^2$ (for details, see Supplementary Note 3). A clear residual $\kappa/T$ in the zero-temperature limit is resolved for H-Cat. The peak structure of $\kappa^H/T$ around 0.3 K is considered to arise from both $\kappa_{sp}^H$ and $\kappa_{ph}^H$ (See Supplementary Note 6)

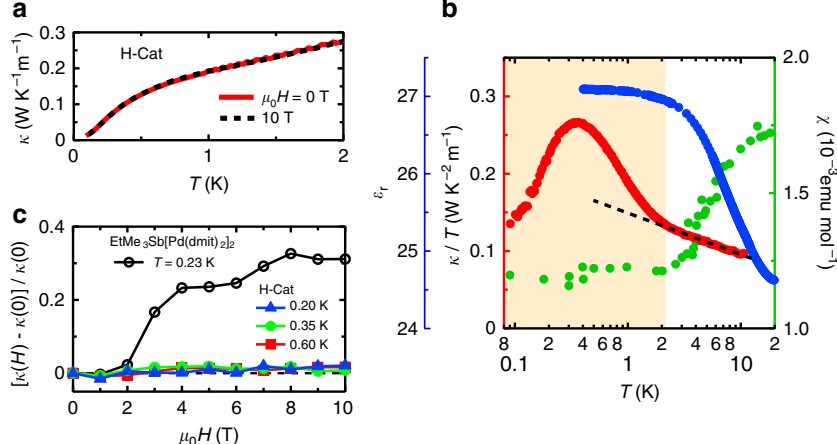

**Fig. 4** Paraelectric and spin liquid phases of H-Cat. **a** Temperature dependences of $\kappa(H)$ for H-Cat at 0 and 10 T. **b** A combination of the temperature dependences of the dielectric constant $\epsilon_r$ (blue, left axis), the thermal conductivity divided by temperature $\kappa/T$ (red, left axis) and the magnetic susceptibility $\chi$ (green, right axis) for H-Cat. The values of $\chi$ are taken from ref. [5]. The dashed line is an eye guide. The shaded region represents the QPE and QSL phases. **c** Magnetic field dependence of the thermal conductivity $\kappa(H)$ normalized by the zero field value $[\kappa(H) - \kappa(0)]/\kappa(0)$ for H-Cat (filled symbols) and EtMe$_3$Sb[Pd(dmit)$_2$]$_2$ (open circles)[26]

This energy scale is close to the proton fluctuations ($T_1 = 7.7$ K), indicating that resonance scattering arises between the acoustic phonons and the optical mode from the hydrogen bonds. Here, it should be noted that we can rule out the possibility of a spin–phonon scattering for the suppression of $\kappa_{ph}^H$, because the spin–orbit coupling of H-Cat is very weak, as confirmed by the small field dependence of $\kappa^H$ (Fig. 4a). Thus, it appears that the thermal fluctuations of the hydrogen bonds strongly suppress $\kappa_{ph}^H$ above 2 K.

Below 2 K, $\kappa^H$ rapidly increases and eventually exceeds $\kappa^D$ (Fig. 3a). This rapid increase of $\kappa^H$ may come from increases in $\kappa_{sp}^H$ as well as $\kappa_{ph}^H$. We now investigate the behavior of $\kappa^H$ at lower temperatures where $\kappa_{sp}^H$ becomes dominant over $\kappa_{ph}^H$; this provides essential information on the low-lying excitation spectrum characterizing the QSL state[25,26]. As shown in the dotted lines in Fig. 3b, both $\kappa_{ph}^H$ and $\kappa_{ph}^D$ exhibit a $T^2$-dependence rather than the conventional $T^3$-dependence. This originates from the influence of high-quality crystals with specular surfaces

(Supplementary Fig. 3; Supplementary Note 3). The zero-temperature extrapolation of $\kappa^H/T$ shows a finite residual (Fig. 3b; Supplementary Fig. 4), thereby demonstrating a gapless spin excitation with high mobility (the mean free path of the gapless spin excitations $l_{sp}$ is estimated to be ~120 nm; See Supplementary Note 4). This result is consistent with recent magnetic torque measurements[5] of H-Cat. Here, we stress that we can exclude the possibility that the itinerant low-energy excitations are due to either phonons or electric dipoles (see Supplementary Note 5).

**Discussion**

A key question raised here is how the gapless QSL state is stabilized in H-Cat. In organic QSL candidates with a triangular lattice charge fluctuations near a Mott transition[12–18] have been pointed out to play an important role for stabilizing the QSLs. However, H-Cat is located deeper inside the Mott-insulating phase compared to the other organic QSL candidates, $\kappa$-(BEDT-TTF)$_2$Cu$_2$(CN)$_3$ (ref. [3]) and EtMe$_3$Sb[Pd(dmit)$_2$]$_2$ (refs. [4,30]). The

distance from a Mott transition is inferred from the ratio of the on-site Coulomb repulsion $U$ to the transfer integral $t$, which is given by $U/t \sim t/J$. Whereas the transfer integrals are comparable among the three materials, $J$ for H-Cat is ~1/3 compared to that for the other two (see Supplementary Note 7); this means that $U/t$ is significantly large in H-Cat. Indeed, H-Cat sustains an insulating behavior even at 1.6 GPa (ref. [20]), whereas the other two compounds become metallic at 0.4–0.6 GPa (ref. [31]); this result also supports that H-Cat is far from the Mott transition. Therefore, the QSL in H-Cat should be stabilized using a different mechanism.

Figure 4b shows the temperature dependence of the dielectric constant $\epsilon_r$, the thermal conductivity divided by temperature $\kappa/T$ and the magnetic susceptibility $\chi$ (ref. [5]) for H-Cat. Below 2 K, the thermal conductivity increases upon entering the QPE state, where $\epsilon_r$ saturates. The characteristic temperature coincides with the temperature at which the susceptibility becomes constant; this occurs when the spin correlation develops in the QSL state[5]. The coincidence of the QPE and QSL states is surprising and strongly suggests that the development of the quantum proton fluctuations triggers the emergence of the QSL. We now theoretically analyze the effects of proton dynamics on the QSL state. In H-Cat, the charge degrees of freedom of the hydrogen bonds and the $\pi$-electrons are strongly coupled because of the charge neutrality within the $H_3(Cat-EDT-TTF)_2$ supramolecule (see Fig. 1c, d). A minimal and realistic model that describes the coupling between $\pi$-electrons and hydrogen bonds is the extended Hubbard model coupled with the proton degree of freedom; this model captures the essence of this system, namely that the proton and charge degrees of freedom are strongly entangled in H-Cat (Supplementary Fig. 5; Supplementary Note 8). According to the present model, charge fluctuations inside a dimer and/or between two neighboring dimers are affected by proton fluctuations for the following reasons: the magnetic exchange processes are governed by the virtual electron hopping (mainly the second term in Supplementary Eq. (7)), which is of the order of 100 meV in the present system[20–22]. In contrast, the time scale of the proton fluctuations $T_1 = 7.7$ K (~1 meV) is two orders of magnitude slower than that of the electron hopping. Such low-energy proton fluctuations modulate the amplitude of the electron transfers and the energy levels of the molecular orbitals. These effects may induce a dynamical modulation of $J$ as well as a reduction of the on-site Coulomb repulsion $U$ due to the bi-polaron effect[32], both of which appear to destabilize the magnetic long-range order, that is, induce a QSL state.

Finally, we discuss the magnetic field dependence of the thermal conductivity in the QSL state of H-Cat. As shown in Fig. 4c, the field dependence of the thermal conductivity at low temperatures in H-Cat is negligibly small (a slight increase of 1–2% against the magnetic field of 10 T). In contrast, in $\kappa$-(BEDT-TTF)$_2$Cu$_2$(CN)$_3$ (ref. [25]) and EtMe$_3$Sb[Pd(dmit)$_2$]$_2$ (ref.[26]), the magnetic field dependence of the thermal conductivity shows a gap-like behavior, which has been discussed in terms of an inhomogeneous QSL[3,25,33–35] and a dichotomy of gapless and gapped excitations[26,30], respectively. Therefore, the observed negligibly small field dependence in H-Cat indicates the absence of gapped excitations with magnetic field, which may suggest a more globally homogeneous QSL with gapless excitations. One of the possible explanations is a gapless spinon Fermi surface over the whole $k$-space[36] (for details, see Supplementary Note 9). Recent torque measurements[5] of H-Cat have shown that spin excitations behave as Pauli-paramagnetic-like low-energy excitations where the Fermi temperature $T_F$ is estimated to be 350 K. In such a case, in the regime where $T_F \gg g\mu_B H$ (here, $\mu_B$ is the Bohr magneton and $g$ is the g-factor of Cat-EDT-TTF dimer with spin-1/2), the total number of spin excitations in the applied magnetic field becomes constant in a 2D system, and the velocity of the spin excitations $v_{sp}$ is assumed to be almost field-independent. As a result, $\kappa_{sp}$ becomes essentially field-independent.

The homogeneous gapless QSL insensitive to magnetic fields may originate from the structure of the present system; the 2D $\pi$-electron layers of H-Cat are connected by hydrogen bonds, whereas in the other organic QSL candidates they are separated by anion layers that may induce randomness in the $\pi$-electron system[37]. This structural difference enables a different mechanism to stabilize the QSL state in H-Cat compared to previous organic QSL candidates. The QSL realized in H-Cat can be induced by quantum proton fluctuations rather than charge fluctuations near a Mott transition. Thus, our findings suggest that a quantum-disordered state of magnetic and electric dipoles emerges in H-Cat from cooperation between the electron and proton degrees of freedom. Utilizing such a strong coupling between multiple degrees of freedom will advance our explorations of quantum phenomena such as orbital–spin liquids[19,38] and electric–dipole liquids[17,39].

## Methods

**Sample preparation.** Single crystals of $\kappa$-H$_3$(Cat-EDT-TTF)$_2$ and $\kappa$-D$_3$(Cat-EDT-TTF)$_2$ were prepared by the electrochemical oxidation method, as described in refs[20,21]. A typical sample size for both compounds is ~0.03 × 0.12 × 1.0 mm$^3$. In H-Cat, the anisotropy parameter $t'/t$ at 50 K is estimated to be 1.48 by the extended Hückel method[5] or 1.25 by the first principles DFT calculations[22] (see Fig. 1e). In contrast, the value of $t'/t$ for D-Cat is estimated to be 1.36 at 270 K by the extended Hückel method[21] (see Fig. 1f), which is close to $t'/t = 1.37$ at 298 K for H-Cat obtained by the same method[40].

**Dielectric measurements.** The dielectric permittivity measurements were carried out down to 0.4 K in a $^3$He cryostat using an LCR meter (Agilent 4980A) operated at 100 Hz–1 MHz along the $a^*$ direction, which is perpendicular to the $b$-$c$ plane. The experiment was limited to the $a^*$-axis direction by the plate-like shape of the sample. The applied a.c. voltage was 2 V. We confirmed the voltage-independent response of $\epsilon_r$ up to 2 V below 20 K. The dielectric permittivity was measured by sweeping both temperature and frequency. The electrical contacts were made using carbon paste. The open/short correction was performed before connecting the sample to the measurement system.

**Thermal conductivity measurements.** The thermal-transport measurements were performed by a standard steady-state heat-flow technique in the temperature range from 0.1 to 10 K using a dilution refrigerator. The heat current was applied along the $c$ axis. The magnetic field was applied perpendicularly to the $b$-$c$ plane up to 10 T. Two RuO$_2$ thermometers precisely calibrated in the magnetic field and one heater were attached on the sample through gold wires.

**Data availability.** The data that support the findings of this study are available on request from the corresponding authors (M.S. or K.H.).

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

## Acknowledgements

We thank J. Müller, M. Oshikawa, H. Seo, T. Shibauchi, M. Tachikawa, Y. Tada, T. Tsumuraya, H. Watanabe, and K. Yamamoto for fruitful discussions. We also thank K. Torizuka and Y. Uwatoko for providing technical assistance. This work was supported by Grants-in-Aid for Scientific Research (Grants Nos. 24340074, 26287070, 26610096, 15H00984, 15H00988, 15H02100, 15K13511, 15K17691, 16H00954, 16H04010, 16K05744, 16K17731, 17H05138, 17H05143, and 17K18746) from MEXT and JSPS, by a Grant-in-Aid for Scientific Research on Innovative Areas "$\pi$-Figuration" (No. 26102001), by the Canon Foundation and by Toray Science Foundation.

## Author contributions

M.S., K.H., and M.Y. conceived the project. M.S., Y.S., K.S., S.Y., Y.I and M.Y. performed the thermal conductivity measurements. K.H., R.K., K.I., S.Iguchi, and T.S. performed the dielectric permittivity measurements. A.U. and H.M. carried out sample preparation. M. N. and S.Ishihara gave the theoretical model. M.S., K.H., and M.Y. analyzed the data and wrote the manuscript with inputs from A.U., M.N. and S.Ishihara. All authors discussed the experimental results.

## Additional information

**Competing interests:** The authors declare no competing financial interests.

