## [Peer Review File · Nature Communications]

Reviewers' comments:

Reviewer #1 (Remarks to the Author):

This manuscript describes dielectric and thermal conductivity behavior of a quantum spin-liquid (QSL) candidate, $H_3(\text{Cat-EDT-TTF})_2$, with the support of theoretical investigations. Overall works were performed well, and especially, concerted phenomenon coupled with QSL and quantum paraelectric behavior through hydrogen dynamics will open new opportunities for exploring quantum phenomena characteristic of organic materials. The present results will significantly contribute to the development of the related fields, and therefore, this paper can be accepted for publication subject to the authors addressing the following comments.

- The authors should draw the molecular structure of $H_2\text{Cat-EDT-TTF}$ in main text. In relation, "Cat-EDT-TTF is catechol-fused ethylenedithiotetrathiafulvalene" on page 3 is better changed to " $H_2\text{Cat-EDT-TTF}$ is catechol-fused ethylenedithiotetrathiafulvalene".
- The authors should describe the t'/t value of H-Cat; 1.48 (extend Huckel method; Phys. Rev. Lett. 2014) or 1.25 (DFT method; Phys. Rev. B 2015) at 50 K. Could you make any comments about the t'/t value of the deuterated salt, D-Cat ?
- 1st paragraph on page 5: Although being the deuterium localization in D-Cat at low temperatures, the authors should provide experimental evidence of antiferroelectric dipole moment in D-Cat. What is the T_0 value of D-Cat ?

Reviewer #2 (Remarks to the Author):

Quantum spin liquids belong to the exotic quantum ground states in which strongly correlated spins residing on the spin frustrated lattices fluctuate and thus prevent an antiferromagnetic ordering even at absolute zero. In 2D, conventional understanding means that the geometric frustration effect is not enough strong and that quantum fluctuations originating from the coupling of spins with charge or orbital degrees of freedom is needed to create QSL. However, this issue is far from being resolved. In this paper, the authors claim to demonstrate that the coupling of localized spins with fluctuations of hydrogen atoms provide a new mechanism which explains the spin liquid state in an organic single-component Mott insulator $\kappa\text{-H}_3(\text{Cat-EDT-TTF})_2$ (shortened as H-Cat). They point out that this mechanism of spin liquid formation differs from previously proposed ones, which were all based on the electrons' internal degrees of freedom only, and which cannot be pertinent for H-Cat system because the system is far from the Mott transition. The claim is novel indeed and of importance to other scientists in community and in the wider field interested to quantum phenomena.

There are two fundamental open questions this paper intends to answer: are the strong quantum proton fluctuations present in H-Cat, and if they are, do they and how do they affect the QSL state. The data are sound, the manuscript is detailed and clearly written, has a good structure, clear conclusions. However, I find that despite the conclusions being sound, the presentation of evidence is not rigorous enough. Therefore, I conclude that the paper deserves to be published only after the following issues and questions have been properly addressed.

In the following I explain what I mean by „the presentation of evidence is not rigorous enough“:

Issue 1. The most interesting idea raised in this paper is that the strong quantum proton fluctuations give rise to a novel QSL. The key result to support such a claim is a dielectric behaviour in accord with the prediction of the Barret formula. This formula describes the behavior of the dielectric constant in the wide temperature range from the low-temperature region where quantum fluctuations are important up to the high-temperature region where a classical approximation and a Curie-Weiss law are valid. In this paper, only the behavior of the real part of

dielectric function at 1 MHz frequency is shown. Data taken at only one frequency do not allow the conclusion that the observed dielectric behavior is valid for the dielectric constant, i.e. that it corresponds to the real part of dielectric function in the zero-frequency limit. In order to support rigorously enough the claim that the paraelectric state is observed, the authors must prove that the dielectric constant behavior is found, i.e. that the dielectric behavior is frequency-independent.

Issue 2. Second important result in this paper is that a rapid rise of the thermal conductivity below 2 K can be ascribed to spinons and that it happens in the same T-region where the dielectric behavior saturates. The important conclusion follows that „this coincidence strongly suggest that the development of the quantum proton fluctuations triggers the emergence of the QSL.“ In addition to that, $\kappa(T)/T$ is found to have a finite residual value implying the presence of gapless excitations. This finding confirms the conjecture already proposed in ref.4. Ref.4 paper reported the gapless QSL, based on SQUID and torque magnetometry, showing no AF order despite high $J \sim 80$ K and a T-independent susceptibility below about 3K.

In order to support that the quantum proton fluctuations and QSL are inherently connected, the authors present the magnetic field influence on the thermal conductivity. The result shown in Fig.4b is very intriguing. I do not see why the authors claim that the small field dependence exists: „As shown in Fig. 4b, the field dependence of $\kappa(H)$ is much weaker in H-Cat than in κ -(BEDT-TTF) $_2$ Cu $_2$ (CN) $_3$ and EtMe $_3$ Sb[Pd(dmit) $_2$] $_2$ “ and “the observed small field dependence of the thermal conductivity in H-Cat...”. Actually looking in Fig.4b, as well as in Inset, this influence is nonexistent. The authors must use clear wording and quantify the magnetic field dependence of the thermal conductivity in a rigorous manner. Commonly, non-existent dependence on magnetic field excludes the possibility that the itinerant low-lying excitations have a magnetic origin, meaning that the thermal conductivity arises either due to phonons or to electric dipoles rather than spinons. The authors should discuss these conventional options. The idea of “a more globally homogeneous QSL” sounds very appealing, but it needs further elaboration.

A more rigorous comment is also needed on the origin of the maximum in thermal conductivity at $H=0$ and how it is related to spinons.

The other questions follow.

Q1. The description of dielectric measurements given in Methods is insufficient: It is not clear how the dielectric permittivity measurements were carried out: the authors need to specify the ac amplitude used, how were the electrodes prepared, the mode of measurement (the temperature-sweep or the frequency-sweep, or both) and how were the extrinsic effects excluded.

Q2. In dielectric measurements, is there any sample dependence and is there any dc bias influence on the dielectric permittivity?

Q3. It is stated that “In H-Cat, $\epsilon_r(T)$ steeply increases with decreasing temperature and then saturates below 2 K.” Is there any reason why dielectric data above 20 K is not shown? It would be certainly interesting to see evolution with cooling. The application of the Barret formula would be better justified if the behavior of dielectric constant in the broader temperature range was covered so that the Curie-Weiss temperature of hydrogen-bond order in the absence of quantum fluctuations could be estimated; is it possible to know down to which temperature, dielectric constant follows the CW law?

Q4: What about the relative strength of quantum fluctuations: how do they compare to other systems or to predictions of DFT calculations?

Q5. A minimal model describing the coupling between π -electrons and hydrogen bonds is suggested. It is stated that „charge fluctuations between the two molecules inside a dimer are

governed by proton fluctuations (≈ 1 meV), which are comparable to the magnetic exchange interaction J in the n -electron spin system." Is it really only valid for two molecules inside single dimer or is it also valid for two molecules, each belonging to a neighboring dimer? It would be useful to specify the value of J and explain in what way the amplitude of proton fluctuations is estimated (from DFT calculations? If yes – cite the paper).

Q6. Elaborate the very strong claim that „Such low-energy charge fluctuations lead to a dynamical modulation of J as well as a reduction of the on-site Coulomb repulsion U due to the bi-polaron effect, both of which may destabilize the magnetic long-range order, that is, induce a QSL state.“

Reviewer #3 (Remarks to the Author):

This is a very interesting paper, describing the role of quantum tunneling of electrical dipoles found within the hydrogen bonds, in affecting the low-energy spin dynamics within the Mott insulating state of organic materials. I find this work really original and quite insightful, because it provides clear new information about the very controversial issue of quantum spin liquid behavior in these systems. I very much like the fact that several complementary experimental probes have been used, allowing insight into how the proton dynamics affects the formation of the spin liquid. This is a unique feature of this material, which was not possible to do in other candidate spin liquid systems, and which will offer a very useful testing ground for various theoretical ideas. Specifically, I like very much the arguments the authors present, suggesting that it is the proton fluctuations within the hydrogen bonds, and not the charge fluctuations expected close to the Mott transition, that play a key role here.

I think that this paper should be published in Nature Communications, provided that the authors provide convincing answers the following physical questions:

1. Is it possible to tune the degree of quantum fluctuations within the hydrogen bonds, by applying pressure in a specific way? The authors mention that pressure has little effect on the system, whereas it is well known that in other systems with quantum tunneling within the hydrogen bonds (e.g. KDP mentioned in the manuscript), the tunneling rate can be significantly affected by pressure. Can this be used to control the present system, and why does pressure seem to have little effect here?
2. In other hydrogen-bonded systems, partial or full deuteration can indeed be used to suppress quantum tunneling of protons, as the authors note. In contrast to what the authors have shown here, *partial* deuteration can also be utilized to control/regulate the overall effect of quantum fluctuations. Is it possible to use this strategy in the present system, and thus examine the gradual crossover between the two regimes they presently describe? Note that interesting effects of partial deuteration on quantum tunneling has been discussed in early theoretical work many years ago (e.g. Phys. Rev. B 37, 3703 (1988)).
3. The theoretical model the authors propose describes the quantum tunneling effect within a given hydrogen bond, and also its coupling to the charge disproportionation within the dimers. In other systems (e.g. in KDP), however, electrostatic and elastic coupling exists between the proton degrees of freedom within the hydrogen bonds, typically leading to ferroelectric (or anti-ferroelectric in ADP) orders, even without the charge disproportionation effect. Why is there no term in the Hamiltonian describing such "direct" ferroelectric interactions between the hydrogen bonds?

====Reply to Reviewer #1

We thank Reviewer #1 for evaluating our manuscript and for the very positive statements with the recommendation of publication in Nature Communications. In the following, we answer the reviewer's comments point by point.

1) We have added a new figure that describes the molecular structure of H₂Cat-EDT-TTF in the revised manuscript (see new Fig. 1b) as suggested by the reviewer. We have also incorporated the reviewer's suggestion that "Cat-EDT-TTF is catechol-fused ethylenedithiotetrathiafulvalene" is better changed to "H₂Cat-EDT-TTF is catechol-fused ethylenedithiotetrathiafulvalene".

2) We have specified the t'/t values of H-Cat and D-Cat in the captions of new Figs. 1e and f, as suggested by the reviewer. Accordingly, we have added a new reference (Ref. 22; A. Ueda *et al.*, Chem. Eur. J. **21**, 15020 (2015)) in the revised main text.

3) In the case of antiferroelectric materials, the net dipole moment of the crystal is zero because the local electric dipoles oriented in antiparallel directions cancel each other out, but one can estimate the local electric dipole moment from the Curie constant $C = n\mu^2/k_B$ obtained from the Curie–Weiss behaviour in dielectric permittivity, where n is the density of dipoles, μ is the local dipole moment, and k_B is the Boltzmann constant. In the case of D-Cat, however, the insulating properties at high temperatures around the transition temperature of 185 K are not good enough to perform dielectric permittivity measurements. Therefore, we are not able to experimentally determine the local electric dipole moment in D-Cat.

T_0 in the Barrett formula represents a temperature at which (anti)ferroelectric order occurs in the absence of strong quantum fluctuations. In the present system, deuteration of the hydrogen bonds suppresses strong quantum proton fluctuations observed in H-Cat, which leads to deuterium localization at 185 K in D-Cat. Therefore, in principle, the T_0 value of D-Cat should be 185 K. However, because the relatively high conductivity (~ 1 S/cm) of D-Cat around 185 K prevents dielectric permittivity measurements as mentioned above, we cannot obtain the T_0 value of D-Cat experimentally.

We have revised the explanation of each parameter in the Barrett formula to make these points clearer (see the first paragraph on page 4).

====Reply to Reviewer #2

We thank Reviewer #2 for evaluating our manuscript and for the very constructive remarks which help us to improve the manuscript. In the following, we address the reviewer's comments in the same order as they appear in the report.

Issue 1.

We have examined the frequency dependence of the dielectric behaviour in H-Cat (100 Hz to 1 MHz) and confirmed that below ~ 10 K the dielectric constant becomes frequency-independent (see new Fig. S1b). The strong frequency dependence of the dielectric constant above ~ 10 K is considered to originate from a large loss tangent due to the relatively high electrical conductivity in H-Cat. Such an extrinsic effect on the dielectric behaviour is often observed in organic conductors with large loss tangents (for example, see Fig. 1 of Ref. 24). Because the loss tangent $\tan\delta$ is given by $1/(\omega CR_p)$, where ω is the angular frequency of the a.c. electric field, C is the lossless capacitance, and R_p is the parasitic resistance, the higher the measuring frequency is, the smaller the dielectric loss becomes. Therefore, in the main text, we have shown the dielectric constant measured at 1 MHz. We have added discussion on this point in new Sec. I in the revised Supplementary Information.

Issue 2.

As pointed out by the reviewer, it is correct that the magnetic field dependence of the thermal conductivity in H-Cat is negligibly small. To make this point clearer, we have quantified the increase in the thermal conductivity against the magnetic field of 10 T (1~2 % at low temperatures). This result may raise a question as to the magnetic origin of the itinerant low-lying excitations observed in the thermal conductivity of H-Cat. However, we can safely exclude the possibility that the itinerant low-energy excitations, which are described by the T -linear term of the thermal conductivity as discussed in the main text, are due to either phonons or electric dipoles for the following reasons: (1) As shown in the dotted lines in Fig. 3b, the temperature dependence of κ/T of both H-Cat and D-Cat exhibits a T -linear dependence at low temperatures. Considering that D-Cat has only phonon contribution to the thermal conductivity, this result indicates that the phonon thermal conductivity in this system is described by a T^2 -dependence (owing to the influence of high-quality crystals with specular surfaces, for details, see Supplementary Information Sec. III) rather than the conventional T^3 -dependence. Because $\kappa_{\text{ph}}/T \propto T$ goes to zero in the zero-temperature limit, the phonon thermal conductivity cannot contribute to the itinerant

low-energy excitation observed in H-Cat. (2) Because electric dipoles behave as classical Ising-like spins, a dielectric system with 1D uniaxial (Ising-type) antiferroelectric dipoles, as in the case of H-Cat, is expected to have low-energy excitations with a gap, which is inconsistent with the gapless excitations in H-Cat. We have added discussion on this point in new Sec. V in the revised Supplementary Information.

The observed negligibly small field dependence of the thermal conductivity in H-Cat rather suggests a globally homogeneous QSL with gapless spin excitations such as a gapless spinon Fermi surface over the whole \mathbf{k} -space (S. S. Lee and P. A. Lee, PRL **95**, 036403 (2005)). Recent torque measurements (Ref. 5) of H-Cat have shown that spin excitations behave as Pauli-paramagnetic-like low-energy excitations with the Fermi temperature T_F of ~ 350 K. In such a case, in the regime where $T_F \gg g\mu_B H$ (here, μ_B is the Bohr magneton and g is the g-factor of Cat-EDT-TTF dimer with spin-1/2), the velocity of the spin excitations v_{sp} is assumed to be almost field-independent. Furthermore, we can assume that the total number of spin excitations in the applied magnetic field becomes constant in a 2D system owing to the cancellation of the spin-up and spin-down parts. As a result, κ_{sp} is essentially field-independent (for details, see new Sec. IX in the revised Supplementary Information). The observed homogeneous gapless QSL insensitive to magnetic field may originate from the unique feature of the hydrogen-bonded molecular conductor H-Cat, where the 2D π -electron layers are connected by hydrogen bonds, not by anion layers, as in the case of the other organic QSL candidates, which may induce randomness in the π -electron (K. Watanabe *et al.*, JPSJ **83**, 034714 (2014)). This structural difference between H-Cat and the previous organic QSL candidates may provide the different mechanisms stabilizing the QSL state. We have added discussion on this point and new references (Refs. 36 and 37) in the last two paragraphs in the revised main text. We have also added the detailed discussion and a related reference (Ref. S20; D. Watanabe *et al.*, Nat. Comm. **3**, 1090 (2012)) in new Sec. IX in the revised Supplementary Information.

The maximum in κ/T of H-Cat could be related to both κ_{sp} and κ_{ph} . In the present system, the spin correlation develops entering the QSL regime as described in the main text; this implies that I_{sp} increases with decreasing temperature below 2 K, followed by the saturation of I_{sp} at low temperatures. In addition, C_{sp}/T has been recently reported to be almost constant with temperature, showing $C_{sp} \propto T$ (S. Yamashita *et al.*, PRB **95**, 184425 (2017).), and v_{sp} is assumed to be essentially temperature-independent at low temperatures. Therefore, κ_{sp}/T below 2 K should be saturated at low temperatures without showing any peak structure. As described in the revised Supplementary Information Sec. III, κ_{ph}/T decreases monotonically with decreasing temperature at least at low temperatures.

Consequently, we conclude that a combination of κ_{sp} and κ_{ph} results in the peak structure observed in κ/T . We have added discussion on this point and a new reference (Ref. S2) in new Sec. VI in the revised Supplementary Information. We have also changed the related sentences in the third paragraph on page 5 in the revised main text.

Q1) As suggested by the reviewer, we have added the details of the experimental conditions for the dielectric permittivity measurements in the Method section as follows: The applied a.c. voltage was 2 V. The dielectric permittivity was measured by sweeping both temperature and frequency. The electrical contacts were made using carbon paste. The open/short correction was performed before connecting the sample to the measurement system.

Q2) We have measured the dielectric permittivity for several samples of H-Cat and confirmed the reproducibility of the observed quantum paraelectric (QPE) behaviour (see new Fig. S1a). We have also examined the d.c. bias dependence of the dielectric permittivity in H-Cat and found that there is no detectable influence of d.c. bias electric fields on the dielectric permittivity (see new Fig. S1a).

Q3) Because the insulating properties of H-Cat become worse at high temperatures, the dielectric constant of H-Cat starts to increase with increasing temperature even at 1 MHz. However, the dielectric constant obtained below 20 K enables us to estimate the Curie–Weiss (CW) temperature T_0 and the crossover temperature T_1 from the classical CW behaviour to the QPE behaviour, as discussed in the first paragraph on page 4. As shown in Fig. 2b, by fitting the data to the Barrett formula, we have obtained $T_0 = -6.4$ K and $T_1 = 7.7$ K.

Q4) The relative strength of quantum fluctuations among different QPE materials can be evaluated by the ratio of T_1 to T_0 . In H-Cat, the T_1/T_0 value is 1.2, while the values in the typical QPE material SrTiO_3 (Ref. 27) is estimated to be 2.3. This is consistent with the experimental fact that the QPE behaviour of SrTiO_3 is more significant than that of H-Cat. We have added this point in the first paragraph on page 4.

Q5) In our theoretical model, we have considered both the charge fluctuations inside a dimer as well as those between two neighbouring dimers. These two kinds of charge fluctuations are represented by the first and second terms, respectively, in Eq. (S7) in

revised Supplemental Information Sec. VIII. We stated in the previous manuscript that charge fluctuations inside a dimer lead to a modulation of J , but it is also true as suggested by the reviewer that charge fluctuations between two neighbouring dimers also induce a modulation of J , especially at the vicinity of the AFE phase observed in D-Cat, where the charge fluctuations between two neighbouring dimers become more remarkable. In order to explain this point more clearly, we have revised a part of the last paragraph on page 6 as “According to the present model, charge fluctuations inside a dimer and/or between two neighbouring dimers are affected by proton fluctuations ...”.

The amplitude of the proton fluctuations corresponds to $T_1 = 7.7$ K (~ 1 meV), which was obtained from the analysis of dielectric permittivity by using the Barrett formula as discussed in the first paragraph on page 4. We have specified the value in the revised manuscript.

Q6) First, we discuss the origin of the dynamical modulation of J . It is known that the magnetic exchange processes are governed by the virtual electron hopping (mainly the second term in Eq. (S7) in revised Supplementary Information Sec. VIII), which is of the order of 100 meV in the present system. This time scale is two orders of magnitude faster than the proton fluctuations ~ 1 meV. Therefore, the displacement of protons largely modifies the electron hopping amplitude and the energy levels of the dimer molecules. As a result, the magnitude of J significantly depends on the proton configuration, that is, the value of J is modulated by the slow proton fluctuations, which we call the dynamical modulation of J in the present manuscript.

The electron-proton interaction also provides a reduction of the Coulomb repulsion U . The attractive (repulsive) interaction between the proton and the electron (hole), which is represented by the second term in Eq. (S8), reduces the energy level of the doubly occupied state of a dimer molecule largely compared to that of the singly occupied state. Such a reduction of the repulsive interaction U between electrons is known as the bipolaron effect due to the electron-ion interaction. Indeed, we have confirmed by numerical calculations that this effect is remarkable in particular near the phase boundary between the Mott insulating and the AFE phases.

Both of the above effects contribute to destabilization of magnetic long-range order. In order to elaborate this point, we have added the above explanation in the last paragraph on page 6: “the magnetic exchange processes are governed by the virtual electron hopping (mainly the second term in Eq. (S7) in Supplementary Information Sec. VIII), which is of the order of 100 meV in the present system¹⁹⁻²¹. In contrast, the time scale of the proton

fluctuations $T_1 = 7.7$ K (~ 1 meV) is two orders of magnitude slower than that of the electron hopping. Such low-energy proton fluctuations modulate the amplitude of the electron transfers and the energy levels of the molecular orbitals. These effects may induce a dynamical modulation of J as well as a reduction of the on-site Coulomb repulsion U due to the bi-polaron effect³², both of which appear to destabilize the magnetic long-range order, that is, induce a QSL state.”

====Reply to Reviewer #3

We thank Reviewer #3 for the very positive statements and for the strong support for publication in Nature Communications. Below we address each comment of the reviewer.

1) It is possible to tune the degree of quantum proton fluctuations by applying pressure in the present system as in the case of KDP, as pointed out by the reviewer. In fact, we have recently observed that the deuterium ordering temperature of D-Cat [REDACTED]

[REDACTED] Also, although according to the previous paper (Ref. 19), H-Cat does not show any pressure-induced phase transition even under 1.6 GPa in the range of 200–300 K (see Figs. 5a and b in Ref. 19), [REDACTED]

[REDACTED] These observations indicate that the pressure application can tune or control the degree of quantum proton fluctuations in this system. Therefore, we emphasize that the pressure effect on the hydrogen-bond dynamics in the present system is not small.

[REDACTED] In the present system, the applying pressure should modulate both the bandwidth of π -electrons and the proton tunneling amplitude. Therefore, we need to consider both the electron and proton degrees of freedom to investigate the pressure effect in the present system. The details of these experiments will be reported elsewhere in the future.

2) It is possible to utilize partial deuteration of hydrogen bonds to control the proton fluctuation strength in the present system. [REDACTED]

[REDACTED] As suggested by the reviewer, partial deuteration of hydrogen bonds in the present system can be a powerful tool to control the coupling strength between the electron and proton degrees of freedom. These results will also be reported elsewhere in the future.

3) We have recognized that in some hydrogen-bonded ferroelectric compounds, direct (anti)ferroelectric interactions arising from electrostatic and elastic couplings between neighbouring protons lead to the (anti)ferroelectric transition, and supposed that such

direct interactions also play some roles in the electronic and structural properties in the present system. In the present theoretical model, however, we focus not on the direct proton-proton interaction, but on the proton-proton interaction through the electron-proton coupling. One of the main reasons is that the experimental results suggest that the hydrogen bonds in the present system strongly couple with the charge and spin degrees of freedom of π -electrons. In addition, there is no direct chemical bond connecting two neighbouring protons. This structural feature causes a screening effect of the electrostatic interaction by the conducting π -electrons, as well as a weak elastic interaction between two neighbouring protons. By taking into account these aspects, we adopted the proton-electron interaction as a main interaction responsible for the observed strong correlation between the electron and proton degrees of freedom. We have added this point in the revised Supplementary Information Sec. VIII.

In conclusion, we believe that we have addressed all the issues raised by the reviewers. With these revisions, we trust that you will find our results suitable for publication in Nature Communications.

REVIEWERS' COMMENTS:

Reviewer #3 (Remarks to the Author):

The authors have, in my opinion, convincingly addressed all the questions I have raised, as well as the questions posed by other referees. I therefore recommend the article for publication.

Reviewer #4 (Remarks to the Author):

This manuscript concerns the quantum spin-liquid candidate material $\text{H}_3(\text{Cat-EDT-TTF})_2$, where experimentally determined dielectric and thermal conductivity properties are supported by theoretical work. The subject matter of quantum proton fluctuations causing a new kind of quantum spin liquid is highly relevant. The revealed physics is sound and the presentation is concise and clear. The authors have answered all questions from previous referees satisfactorily and in detail. As a result, the current version of the manuscript reads very well.

I have one minor comment. Under Methods, the reported amplitude of the applied a.c. voltage in dielectric measurements is 2V. This value is typically considered large and may raise concerns about linearity of response. Supplementary Information does in fact show that d.c. bias has no effect, which implies the a.c. measurements at 2V are well within the linear regime, but this information needs to be looked for and interpreted from another document. I recommend the authors put a short comment under section Methods on the linearity of a.c. response at 2V. This would be enough to put to rest any possible doubts that might emerge on the quality of dielectric data.

With this small addition I can fully recommend the manuscript to be published in Nature Communications without further review.